# Live Weight Prediction of Cattle Based on Deep Regression of RGB-D Images

Alexey Ruchay [1,2,*], Vitaly Kober [2,3], Konstantin Dorofeev [1], Vladimir Kolpakov [1,4], Alexey Gladkov [2] and Hao Guo [5]

1   Federal Research Centre of Biological Systems and Agro-Technologies of the Russian Academy of Sciences, 460000 Orenburg, Russia
2   Department of Mathematics, Chelyabinsk State University, 454001 Chelyabinsk, Russia
3   Center of Scientific Research and Higher Education of Ensenada, Ensenada 22860, Mexico
4   Department of Biotechnology of Animal Raw Materials and Aquaculture, Orenburg State University, 460000 Orenburg, Russia
5   College of Land Science and Technology, China Agricultural University, Beijing 100083, China
*   Correspondence: ran@csu.ru

**Abstract:** Predicting the live weight of cattle helps us monitor the health of animals, conduct genetic selection, and determine the optimal timing of slaughter. On large farms, accurate and expensive industrial scales are used to measure live weight. However, a promising alternative is to estimate live weight using morphometric measurements of livestock and then apply regression equations relating such measurements to live weight. Manual measurements on animals using a tape measure are time-consuming and stressful for the animals. Therefore, computer vision technologies are now increasingly used for non-contact morphometric measurements. The paper proposes a new model for predicting live weight based on augmenting three-dimensional clouds in the form of flat projections and image regression with deep learning. It is shown that on real datasets, the accuracy of weight measurement using the proposed model reaches 91.6%. We also discuss the potential applicability of the proposed approach to animal husbandry.

**Keywords:** live body weight; prediction; image regression; cattle; deep learning

## 1. Introduction

More than one billion head of livestock are reviewed annually worldwide for their breeding and commercial value, health, and prospects for use. At the same time, most measurements and expert assessments are time-consuming and subjective. Live weight is an important factor in animal productivity, providing an informative indicator for feeding, health, breeding, and selection of livestock. In addition, the measurement of animal body weight is one of the most important production tools available to farmers, playing an important role in nutrition, productivity, health, and marketing [1].

Currently, there are two main approaches to measuring body weight [2], that is, the use of industrial scales and indirect methods based on the relationship between body morphological parameters and body weight. Manual measurement of animal body size is time-consuming, labor-intensive, and expensive. Note that simply manual weighing an animal under stress results in a 5–10% reduction in weight and productivity. In addition, it is stressful for both the worker and the animal. Recently, non-contact estimation of morphometric dimensions using low-cost sensors and machine vision methods has been developed [3,4]. The use of non-contact technology significantly reduces the time spent on manual and subjective grading to predict the live weight of livestock or direct weighing of livestock using scales.

Body measurements are commonly used to predict the live weight of animals [5–7]. At the same time, to accurately predict live weight, body size measurements can be used along

with other parameters characterizing the animal: age, sex, body condition assessment, genotype, body volume, body area, etc. Most recent studies have used multiple linear regression analysis to predict body weight. However, these traditional methods are inadequate for accurate prediction [8]. Recently, various machine learning algorithms for live weight prediction using animal morphology have been successfully applied [9–12]. These studies have shown the potential of machine learning algorithms to accurately predict the non-linear relationship between body weight and animal morphology [8]. Moreover, live weight prediction can be based on automatically measured morphological traits using 2D vision systems [12,13] and 3D vision systems [1,4]. The authors of [14] proposed a system for estimating the body weight of a dairy cow with an error of 5.2% based on three linear measurements made using a 3D camera. However, a common disadvantage of such systems is that the accuracy of weight estimation depends on many factors: the prediction model, the quality of measuring morphological features, the choice of model input variables, and insufficient evidence of effectiveness due to a small data sample.

Another promising approach to live weight prediction is to develop a model based on image regression. Image regression is a widely used task in computer vision to predict age, head posture, and facial key points [15]. The easiest way to predict the live weight of cattle is to use RGB images and depth maps, point clouds, or reconstructed dense 3D whole models [16,17]. The choice of the location of the cameras used to capture images of the animal is also crucial. The side view of the animal provides more information but is technologically more difficult due to the requirement to strengthen and clean the cameras. The top view is more acceptable in real farm conditions, as there are no such restrictions.

In a preliminary study [17], the proposed MRGBDM model with a *MAPE* of 9.1% using entire RGB images and depth maps yields the best performance. In this study, only animal areas are selected on RGB-images. Then, 3D augmentation of color projections and 2.5D depth maps by means of rigid transformations in the form of three-dimensional rotation, scaling, and translation is exploited to increase the accuracy of predicting the body weight.

The objective of this work is to develop a reliable model for predicting body weight based on image regression using deep learning methods. For image regression, RGB images and side view depth maps are used to predict the live weight of cattle. The advantage of using only one view, such as side view, is as follows: no need to synchronize data between several sensors, no need to perform complex labor-intensive procedures for external camera calibration, no need to reconstruct a dense three-dimensional animal model, and the ability to use only one camera, which will reduce the cost of contactless technology measurement of the live weight of an animal.

Neural network training requires high image quality. RGB images have a resolution of $1920 \times 1080$ and a depth map of $512 \times 424$, which may distort or inaccurately describe the characteristics of the object. First, RGB images and depth maps from the RGB-D sensor are filtered to improve their quality [18]. The depth map is described by piecewise-smooth areas bounded by sharp object boundaries, so the depth value changes abruptly, and a small error around the object boundary can lead to significant artifacts and distortions. Additionally, the depth map is noisy due to infrared reflections, and missing pixels without any depth value look like black holes in the depth maps. Median and binomial filters [18] are used to reduce noise and fill small holes. Noise and holes affect the accuracy of live weight prediction based on image regression, so denoising and hole filling algorithms need to be used for the live weight prediction model.

The input to a deep neural network can be 2D RGB images or depth maps. However, deep neural networks with point cloud input can be explored in the future. With a limited number of images available, there is little variability in the data, which can lead to overfitting. Moreover, since the sample set for training the neural network is quite small, it is necessary to supplement the training data with synthesized and modified images. There are two options for augmentation; that is, to complement raw 2D RGB images and depth maps, and to project the point clouds obtained from the depth map onto the 2D image plane with an orthogonal projection called a 2.5D depth map. The latter is a more complex and better

way to provide higher variability and similarity of modifications to reality. Previously, the background is removed from the point cloud by extracting the scene from the frame, the alignment of the animal's pose, and then rigid transformations are added in the form of three-dimensional rotations, scaling, and translation. The point cloud is projected onto both the color components (color projection) and the depth map (2.5D depth maps).

Three models for live weight prediction of cattle based on RGB-D image regression and deep learning are proposed. The best model yields a high weight measurement accuracy of 91.6%. The main contribution of this work is as follows:

- Efficient preprocessing of RGB images and depth maps, as well as creating a color RGB projection and 2.5D depth map for subsequent live weight prediction based on image regression with deep learning, are proposed;
- A method for 3D augmentation of color projection and 2.5D depth map using rigid transformations in the form of three-dimensional rotations, scaling, and translation is proposed, which significantly increases the limited dataset and improves the efficiency of live weight prediction in the presence of variations in the posture and position of the animal;
- An efficient model for predicting live weight based on image regression with deep learning is proposed.

## 2. Related Works

In [2], four modeling approaches were identified for predicting the live weight of cattle with different levels of complexity. For all models, three main components can be distinguished, such as feature extraction, feature selection for modeling, and learning model. All components can be automated.

First approach This is a traditional approach in which preliminary models for predicting body weight are based on manual collection of morphometric measurements [19,20]. Some of the most informative morphometric measurements include chest girth, height at withers, hip width/height, and body length. These dimensions are manually selected and used as features for traditional regression models, resulting in single or multiple variable prediction equations based on the number of selected dimensions across species. However, this manual measurement of animal body size is time-consuming, laborious, and expensive, which can lead to a loss of 5–10% of the animal's weight and productivity due to stress.

Second approach To reduce stress on animals and significant costs associated with the traditional approach, the second approach (the CV approach) uses CV systems and images acquired with 2D (RGB and thermal imaging cameras) or 3D (depth and Microsoft Kinect sensors, stereo cameras) electro-optical sensors as an alternative way to capture morphometric measurements. This approach includes an additional step of manual or automated preprocessing of acquired images and manual selection of animal biometric and morphometric measurements, which are then used as predictor variables in statistical models to predict body weight. When 2D images are acquired from only one camera, there is no third dimension, which limits the choice of morphometric measurements for modeling. For example, the HG circumference is reduced to the HG diameter or replaced by chest depth measurements when extrapolating side or top view images [21]. This limitation can be overcome with 3D cameras, but their excessive cost and more complex data processing steps represent the current bottleneck for wider adoption. Alternatively, 2D images can be used for morphometric measurements of perimeter and area, which are features in the model and cannot be easily assessed with manual measurements. In [21], digital image analysis was shown to be reliable in assessing the live weight of female Holstein calves. Compared to traditional measurement methods, the use of digital image analysis reduces the costs, risks for employers, and animal stress associated with measuring and weighing calves.

Third approach Since feature selection can be a complex task, especially with many morphometric measurements, it is preferable to automate this process. Thus, the third approach (the CV + ML approach) includes systems using CV methods as described in the CV approach and machine learning (ML) methods to automate feature selection [12,22,23].

Both the CV approach and the CV + ML approach involve some manual operations such as image and feature selection, image segmentation, and extraction of morphometric measurements. Since this approach uses manual operations, full automation is impossible.

The authors of [24] propose a weighing system for broiler chickens based on a 3D camera. However, further development of broiler segmentation is required, as poor segmentation results in poor weight prediction. Improved segmentation can also solve the problem of tracking broilers, which can lead to a better prediction of individual broiler weight by progressively refining weight estimates across multiple images. In the study [25], a new robust feature extraction method within the V3D computer vision system was developed and applied to automatically estimate the height and body weight of heifers. The authors of [1] used a low-cost depth camera (Microsoft Kinect v1) for non-contact extraction of pig body dimensions and subsequent weight estimation based on two possible models (linear regression and regression of the second degree).

The authors of [26] proposed an automatic weight prediction system for Korean cattle using the Bayesian ridge algorithm on an RGB-D image. This system consists of segmentation, extraction of features, and estimation of the weight of Korean cattle. The prediction system is based on the Bayesian ridge algorithm, rather than on RGB-D image regression.

Fourth approach The fourth approach based on CV and DL (the CV + DL approach) represents the first step towards fully automating the body weight prediction process using digital images. The DL modeling component typically includes image selection, morphometric feature extraction, and selection as part of complex neural network architectures such as Convolutional Neural Networks (CNN), Recurrent Convolutional Neural Networks (RCNN/RNN), and Recurrent Attention Models (RAM). Preliminary animal studies using this approach have shown significant improvements in body weight prediction over more traditional approaches. However, there is still a room for improvement, especially in the accurate automatic segmentation of animals in images with complex, mixed backgrounds with a similar color to objects or several objects.

The authors of [27] proposed a structure-from-motion (SfM) photogrammetry method as a non-invasive and low-cost approach to 3D reconstruction of the pig body. The authors of [28] developed preprocessing algorithms including instance segmentation, distance independence, denoising, and rotation correction. They can save pig body information and remove environmental influences. After processing, the images are passed to the weight prediction model. Their weight prediction model was developed in BotNet. The authors of the paper [29] show that convolutional neural networks do a good job of calculating weight in 2D images. However, incorrect data can greatly reduce the accuracy of the system. While the authors have achieved a much lower error rate than models trained on hand-selected features, there is still more work to be done to eliminate the large errors that can result from these invalid data points, as they can occur when the model is implemented in practice. The authors of [30] proposed a method for estimating the weight of pigs from images without restrictions on pose and lighting. The pig weight estimation method without restrictions on pig posture and image capture environment uses only 2D features for the weight estimation model and the latest advances in machine learning. New features were introduced to describe posture: curvature and deviation. However, the original images may have problems with multiple pigs or only part of the body. The authors of [31] presented a fat measurement system for Angus cows and bulls that uses curvature to describe body shape. The authors of [32] illustrate the success of using 3D cameras to assess body weight and milk properties by measuring the back characteristics of dairy cows.

## 3. Materials and Methods

### 3.1. Datasets

Experiments are carried out on two open datasets from [3]. The first dataset contains the RGB-D data, weights, and manual measurements of 154 Hereford cattle. The Hereford cattle belonged to a private farm with concentrated feed at the age of 12 to 15 months. The animals weighed between 243 and 605 kg. The dataset was collected by an RGB-D imaging

system consisting of two Microsoft Kinect v2 cameras. Two RGB-D cameras are located on the right and left sides of the animal passage at a distance of about 2 m from the animal. Figure 1 shows RGB images and depth maps of cattle taken with two Microsoft Kinect cameras. In experiments, RGB images and depth maps were used on the right and left sides separately. The complete dataset consists of 5220 RGB images and 5220 depth maps on the right side, and 4620 RGB images and 4620 depth maps on the left side.

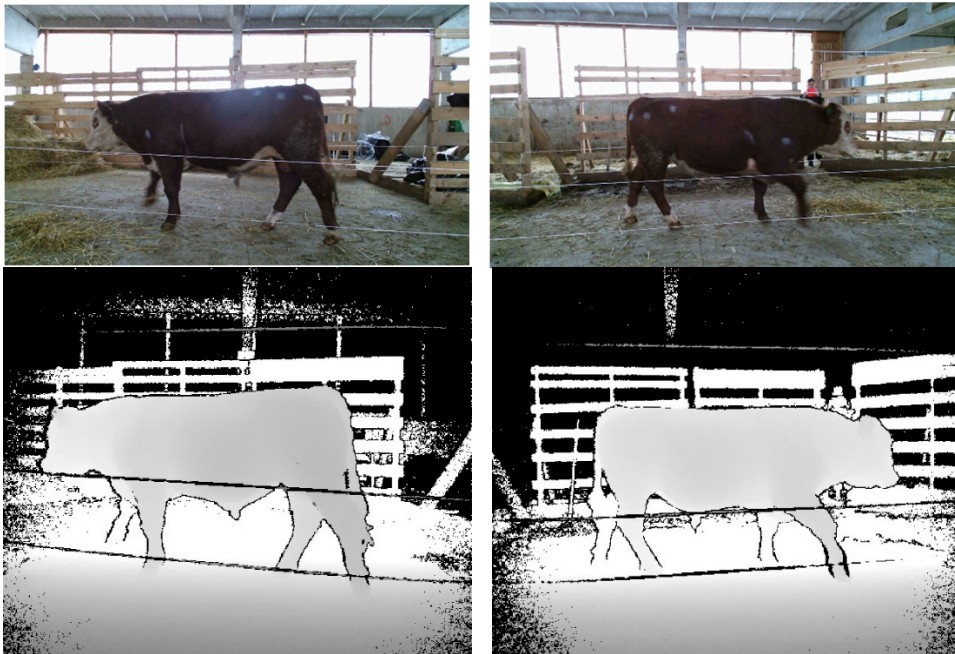

**Figure 1.** RGB images and depth maps of the Hereford cattle.

The second dataset consists of 121 young Aberdeen Angus cattle [33]. At the time of the experiments, the average age of the animals was 16.5 months, and the average body weight was 615 kg. The dataset contains the following data: RFID chip number, RGB images, depth maps and point clouds, and live weight. This database is in the public domain [34]. Two Microsoft Kinect v2 cameras are located on the right and left sides of the animal passage at a distance of about 2 m from the walking animal. Each depth camera was connected to a laptop, and all laptops were connected to a local network. The synchronously acquired RGB-D images were recorded on the respective laptop for each camera. Data collection and storage were implemented based on the Kinect v2 SDK. Each camera was initialized with a trigger signal and started capturing frames at a rate of 30 Hz. The time on the laptops was synchronized, and the best point cloud match could be chosen within the shortest time intervals between the three devices. The resolution of RGB images and depth images is $1920 \times 1080$ and $512 \times 424$ pixels, respectively. Figure 2 shows RGB images and depth maps of cattle taken with two Kinect cameras. The complete dataset consists of 4180 RGB images and 4180 depth maps on the right side, and 3860 RGB images and 3860 depth maps on the left side.

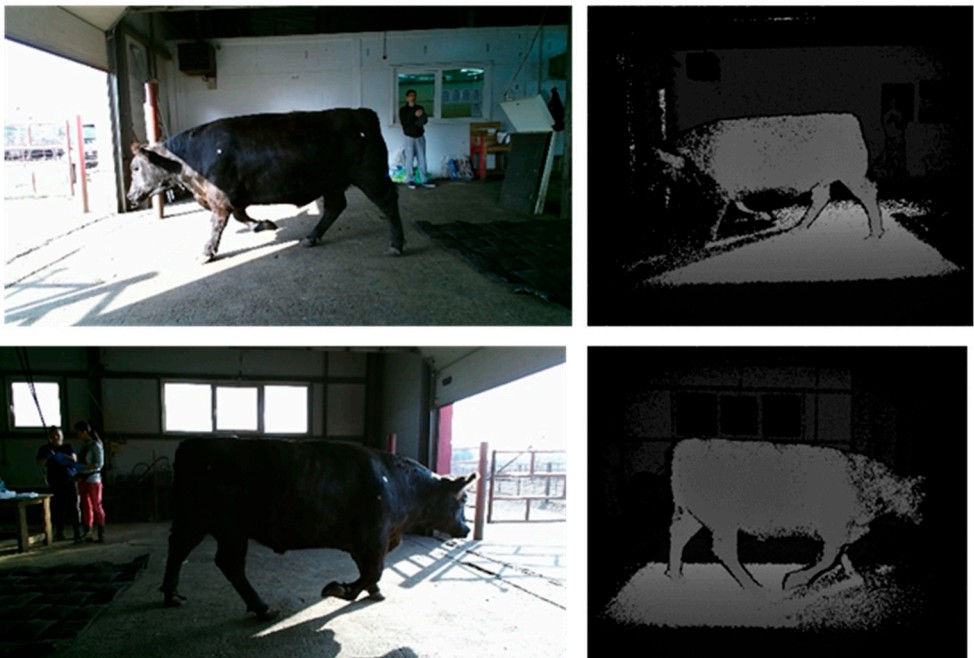

**Figure 2.** RGB images and depth maps of the Aberdeen Angus cattle.

### 3.2. Preprocessing of Data

Due to the high resolution RGB image, the detection result of the target area in the image is more reliable than the result of 3D point cloud detection. Therefore, the existing detection model YOLO v4 [35] was used to detect several regions of different sizes in one 2D image, and the model was further trained to detect areas of the head, thigh, and body of an animal in a 2D image. The presence of three areas of the body, thigh, and head of the animal uniquely defines the frame for the whole animal. As shown in Figure 3, regions are identified with three colors, and the detected regions are represented by a 2D window. Thus, all the initial data were processed to select only frames with whole animals.

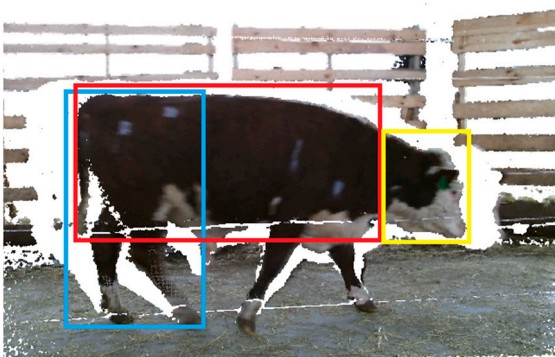

**Figure 3.** Areas of the body, thigh, and head of the animal are marked in red, blue, and yellow, respectively.

The accuracy of the measurement and prediction of live weight can be affected by the posture of the animal. According to [35], the requirements for correct posture can be summarized as follows: the four hooves of the animal being measured must form a rectangle, and the trunk branch must be an almost straight line. The authors of [36] proposed a skeleton extraction method that provides a better way to estimate posture for subsequent body weight prediction. Because body weight estimation methods can be susceptible to incorrect animal postures, a posture selection scheme must be defined to ensure that the correct posture is selected for subsequent measurements. In the data sequence, we consider only those frames on which all the legs of the animal are present.

The first processed dataset for 154 animals consists of 1701 RGB images and 1701 depth maps on the right side, and 1406 RGB images and 1406 depth maps on the left side. The second processed dataset for 121 animals consists of 1536 RGB images and 1536 depth maps on the right side, and 1327 RGB images and 1327 depth maps on the left side.

### 3.3. Denoising of Data

Impulse noise often occurs in color digital images due to sensor malfunctions, transmission errors, and analog-to-digital conversion errors. To improve the quality of color images, it is important to use efficient approaches to estimate distortion parameters and then remove impulse noise. Noise on color images is removed using morphological filtering, where damaged pixels are detected and then removed using morphological filtering [18].

The depth map is described by piecewise smooth areas bounded by sharp object boundaries. Therefore, the depth value changes abruptly, and a small error around the edge of the object can lead to significant artifacts and distortions. Additionally, the depth map is noisy due to infrared reflections, and missing pixels without any depth value look like black holes in the depth maps. Noise and holes can affect the accuracy of body weight prediction, so noise reduction and hole filling algorithms must be used. A switchable two-sided filter to remove noise from the RGB-D depth map is utilized [37]. Bilateral filtering switching is applied only to those pixels of the depth map that are at the edges and show abrupt changes in the signal. First, regions with sharp changes and edges in the RGB image are detected, and then filtering is applied only to the corresponding regions in the depth map.

Recently, many methods have been proposed for filtering 3D point clouds. The ROR algorithm implemented in PCL [38] gives the best result in terms of point cloud reconstruction accuracy using the Hausdorff metric among existing algorithms. ROR removes outliers well if the number of neighbors in a certain search radius is less than a given threshold. An example of a processed point cloud with a threshold for the number of neighbors equal to 20 within a radius of 0.05 is shown in Figure 4.

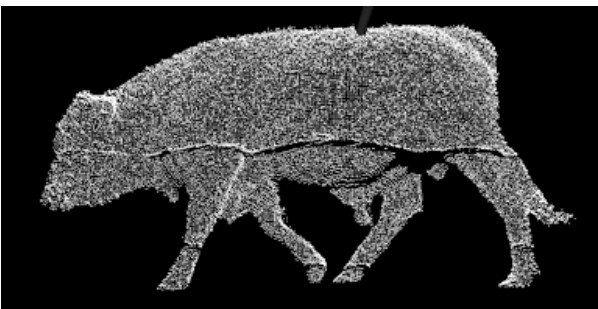

**Figure 4.** Filtered point cloud.

### 3.4. Removing the Background from a Point Cloud

An important step in point cloud processing is to remove the background from the point cloud. Two point clouds are needed, that is, one cloud with a frame without an animal, and the other cloud with the animal. To do this, the entire sequence of frames obtained during recording is used to find a frame without an animal. Next, a point-by-point comparison of cloud points is performed. If the distance of the corresponding points in these clouds in three-dimensional space exceeds a specified threshold value, then the point is transferred to a new point cloud without background. An example of how the algorithm works with the threshold equal to 0.1 is shown in Figure 5.

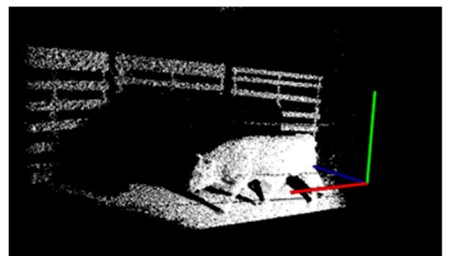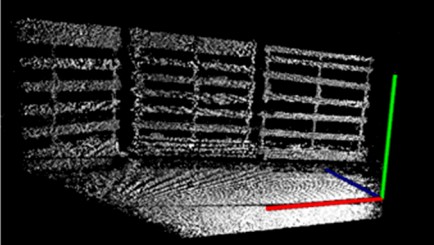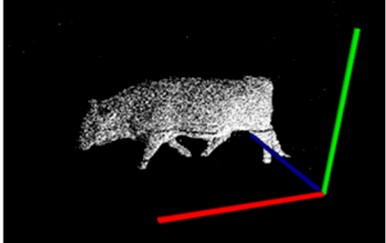

**Figure 5.** Point cloud with the animal, point cloud with the background, and the resultant cloud after removing the background.

### 3.5. Pose Normalization and Lines of Symmetry Calculation

Bilateral symmetry is an important and universal concept for describing animals. The symmetrical plane of the animal is used to obtain the X, Y, and Z axes. Finding suitable orientations often helps in the automated search and processing of 3D objects. In addition, pose normalization helps machine learning algorithms take into account pose information, making object recognition predictions more accurate. A fast bilateral symmetry detection algorithm for a point cloud is proposed. First, the principal component analysis (PCA) algorithm detects the initial symmetry. Then, by exhaustive search of the symmetry planes passing through the center of gravity relative to the initial symmetry plane, the optimal symmetry plane in terms of the modified Hausdorff metric is found. The proposed method consists of the following steps:

1.  Construction of an axis-aligned box bounding the animal in the point cloud. The algorithm is implemented in PCL, and it is equivalent to taking minimum/maximum values at each coordinate of the point cloud;
2.  Place the origin of the coordinate system at the center of gravity of the point cloud;
3.  Estimation of the initial symmetric plane $ax + by + cz = 0$ using the PCA algorithm;
4.  The covariance matrix of the point cloud is calculated, and its eigenvalues and normalized eigenvectors are obtained;
5.  Calculation of the center of gravity $\left(g_x, g_y, g_z\right)$ as follows:

$$g_x = \frac{1}{n}\sum_{i=1}^{n} p_x^i, g_y = \frac{1}{n}\sum_{i=1}^{n} p_y^i, g_z = \frac{1}{n}\sum_{i=1}^{n} p_z^i \qquad (1)$$

where $\left\{\left(p_x^i, p_y^i, p_z^i\right)\right\}, i = 1,\ldots, n$ is the point from the point cloud, and where $n$ is the number of the point cloud $C$;

6.  An exhaustive search of symmetry planes passing through the center of gravity $\left(g_x, g_y, g_z\right)$ relative to the initial symmetry plane in order to find the optimal symmetry plane in terms of the modified Hausdorff metric:

    (a)   splitting the point cloud into two smaller clouds $C_R$ and $C_L$ with the help of the initial symmetry plane $ax + by + cz = 0$ as follows:

$$\left\{\begin{array}{l} p \in C_R, \; ap_x + bp_y + cp_z \leq 0, \\ p \in C_L, \; ap_x + bp_y + cp_z > 0 \end{array}\right\}, \qquad (2)$$

    where $p = \left(p_x, p_y, p_z\right)$ is the point of the point cloud;

    (b)   construction of the mirror reflection $C\prime_R$ of the point cloud $C_R$ as follows:

$$p\prime_x = \left(1 - 2a^2\right)p_x - (2ab)p_y - (2ac)p_z, \qquad (3)$$

$$p\prime_y = \left(1 - 2b^2\right)p_y - (2ab)p_x - (2bc)p_z, \qquad (4)$$

$$p\prime_y = \left(1 - 2b^2\right)p_y - (2ab)p_x - (2bc)p_z, \qquad (5)$$

where $(p_x, p_y, p_z)$ is the point of the cloud $C_R$, and $(p\prime_x, p\prime_y, p\prime_z)$ is the corresponding point of the cloud $C\prime_R$;

(c)　calculation of the Hausdorff metric $d_H$ between $C\prime_R$ and $C_L$ using the average distance as follows:

$$d_H(C_R, C_L) = \max\left(\frac{1}{|C\prime_R|}\sum_{x \in C\prime_R}\min_{y \in C_L}d(x,y), \frac{1}{|C_L|}\sum_{y \in C_L}\min_{x \in C\prime_R}d(x,y)\right), \quad (6)$$

where $|C\prime_R|$ and $|C_L|$ are the number of points in the clouds $C\prime_R$ and $C_L$, respectively, and $d$ is the Euclidean metric.

The first step requires $n$ operations to split the cloud into two smaller clouds. As a result, there should be approximately $\frac{1}{2}n$ points in each resulting cloud. The complexity of calculating the metric is $O(n^2)$ because each element from one cloud is taken and the distance between it and all the elements of another cloud is calculated. Hence, the algorithm has complexity $O(n^2)$.

The results of the PCA are good enough only for the $X$ axis. In the case of the $Y$ and $Z$ axes, the deviations are quite noticeable but lie within fairly small limits, say 0.1 from the coordinates of the axes. Therefore, in order not to try all the planes, one can take the planes obtained as a result of the PCA and slightly change the values of a, b, and c in their plane equations. The correctness of the algorithm can be assessed visually by looking at the object.

To evaluate the accuracy of the proposed symmetry detection algorithm on real data, the proposed algorithm is compared with the PCA algorithm. Symmetry planes along the $X$, $Y$, and $Z$ axes for the PCA algorithm are displayed as a red-colored plane, and for the proposed algorithm they are displayed as a green-colored plane. The results of symmetry detection on real data are shown in Figure 6. For the scanned animal model, the symmetry plane along the $X$ axis from the PCA algorithm coincides with the proposed algorithm. For the $Y$ and $Z$ axes, the proposed algorithm slightly corrects the errors.

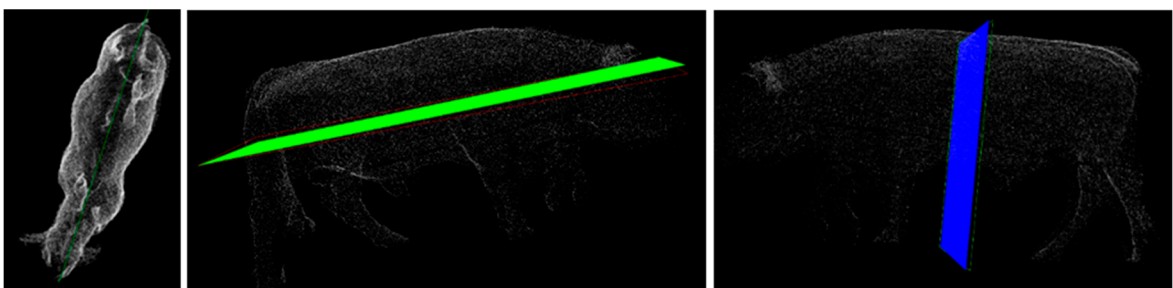

**Figure 6.** Planes of symmetry of a cow.

Symmetry lines are calculated for the entire cloud using the proposed method. Further, to align the cloud, using the found coefficients of the plane and the Cartesian basis of its own subspace, the pose of the animal is aligned parallel to the normalized plane OXZ. Figure 7 shows an example of an aligned point cloud.

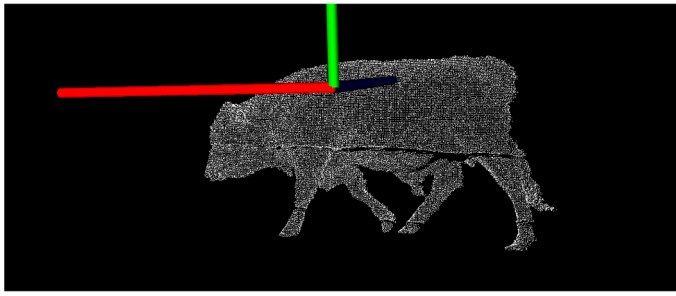

**Figure 7.** Aligned point cloud of the animal.

### 3.6. Calculation of Depth Map Projection (2.5D Depth Map)

The following algorithm is proposed to calculate flat projections of point clouds. The cloud obtained at the previous step is transferred to the origin of coordinates (i.e., the extreme point of the parallelepiped into which the cloud is inscribed moves to the point (0, 0, 0)). Using the OpenCV library, an empty image of $299 \times 150$ is created in memory. The maximum parameters for the width and height of the animal are 2.5 and 1.35 m, respectively. All point coordinates of the cloud are normalized to the size of the image, and the x and y coordinates of the point from the cloud in the empty image are set to the z color value. The resulting image is saved in BMP file format. Figure 8 shows an example of such an alignment.

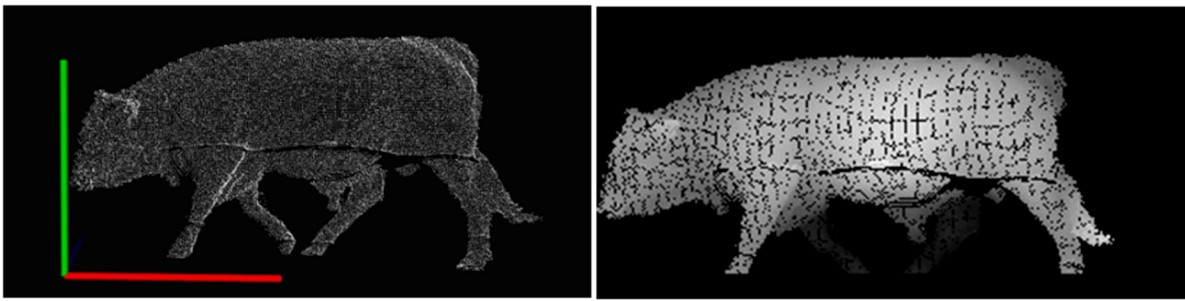

**Figure 8.** Example of the resulting depth map projection (2.5D depth map).

### 3.7. Color Projection

Since the clouds are initially very sparse (the number of points in them is relatively small), their color projections have large distances between pixels. A simple increase in the number of points in the cloud does not lead to an acceptable result, so an algorithm is proposed for calculating the color projection. The algorithm is based on median filtering in each window. Therefore, the window radius is set, the central pixel is replaced by the average value of all pixels that fall into the window, while the replacement condition is that there are at least k pixels in the square, because a small number of pixels gives an unacceptable result. As a result of numerous experiments, the window size of 7 gives an acceptable result, and the value 13 was chosen as k. The complexity of the proposed algorithm for calculating the color projection is estimated as $O\left(\frac{nm}{s}\right)$, where n and m are the width and height of the image, and s is the area of the window. Figure 9 shows an example of the color projection.

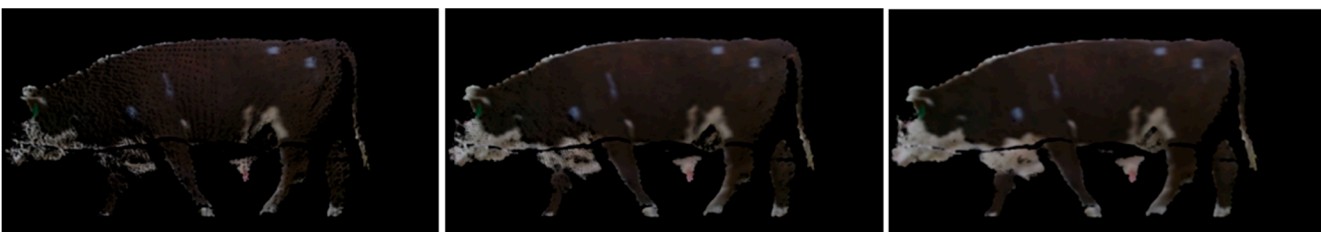

**Figure 9.** Example of color projection for a window size of 5, 6, and 7, respectively.

### 3.8. Image Preprocessing for Neural Networks

3.8.1. Image Resize

The depth map resolution is $512 \times 424$ pixels and the RGB image resolution is $1920 \times 1080$ pixels, so the image needs to be resized to the aspect ratio defined as r = w/h, where r is the aspect ratio, and w and h are the width and height of the image, respectively. Aspect ratio is considered for image resizing as a parameter that helps preserve the best quality of the original image in the downscaling and upscaling procedures. Most of the pre-trained models use an image size of $224 \times 224$ (width, height), which is used as the

target size. When increasing the image size, cubic interpolation is usually used, and when decreasing the image size, area interpolation gives the best results.

### 3.8.2. Signal Range Normalization

After image preprocessing, depth maps and RGB images can have different ranges of pixel values. Therefore, before training the model, the data must be normalized to ensure the same range of pixel values. There are three main pixel value scaling methods supported by the ImageDataGenerator class from the Keras library: pixel normalization (scaling pixel values to the range of 0–1), pixel centering (scaling pixel values to zero mean), and pixel standardization (scaling pixel values to zero mean and unit variance). The pixel standardization algorithm achieves the best performance. The StandardScaler normalization algorithm is based on removing the mean and scaling to unit variance, and is defined as:

$$v = \frac{x - u}{s}, \tag{7}$$

where $x$ represents the current feature value, and $n$ is the normalized feature value, $u$ is the mean of the training samples, and $s$ is the standard deviation of the training samples.

### 3.9. Deep Learning Models

Deep learning is a general machine learning technique in which a model is trained without specialized algorithms for specific problems but uses hierarchical or layered learning. Currently, the Convolutional Neural Network (CNN) is the most popular architecture used in computer vision. Deep learning from the Keras library is used.

The model (MRGBDM) for predicting cow live weight is shown in Figure 10, where inputs are RGB images and depth maps, or a color projection or a 2.5D projection. The model contains three convolution blocks (con1, conv2, and conv3) and two fully connected layers (FC and OUT). The conv1 block has two layers with 64, $3 \times 3$ filters, the conv2 block has two layers with 128, $3 \times 3$ filters, and the conv3 block has three layers with 256, $3 \times 3$ filters. The conv1, conv2, and conv3 blocks are followed by subsampling layers with a kernel size of $2 \times 2$. The Rectified Linear Unit (ReLU), which is not shown in Figure 10, is the activation function applied after each convolutional layer and fully connected layer.

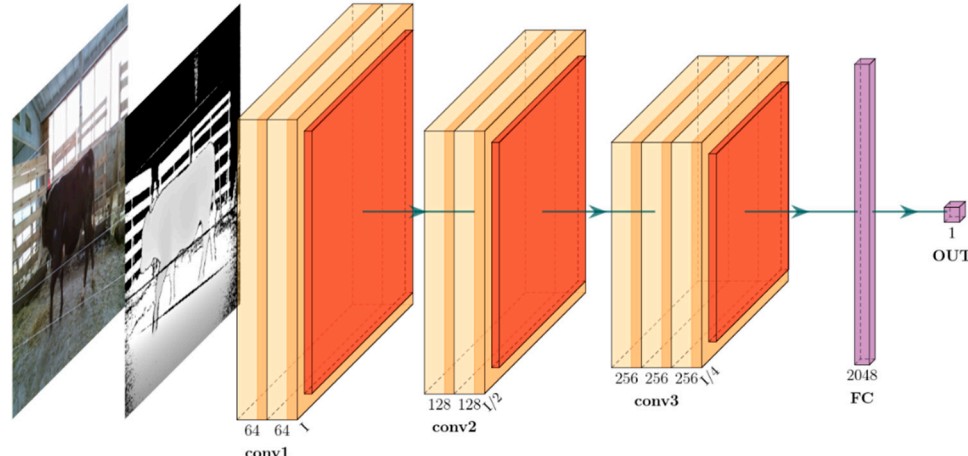

**Figure 10.** MRGBDM Convolutional Neural Network.

The model (MRGB) for cow live weight prediction is shown in Figure 11, where only RGB images are input. The model contains 3 convolution blocks (con1, conv2, and conv3) and two fully connected layers (FC and OUT). The conv1 block has 32, $3 \times 3$ filters, the conv2 block has 64, $3 \times 3$ filters, and the last convolution block, conv3, has 128 $3 \times 3$ filters. After blocks conv1, conv2, and conv3, subsampling layers with a kernel size of $2 \times 2$ are used. ReLU, which is not shown in Figure 11, is the activation function applied after each convolutional layer and fully connected layer.

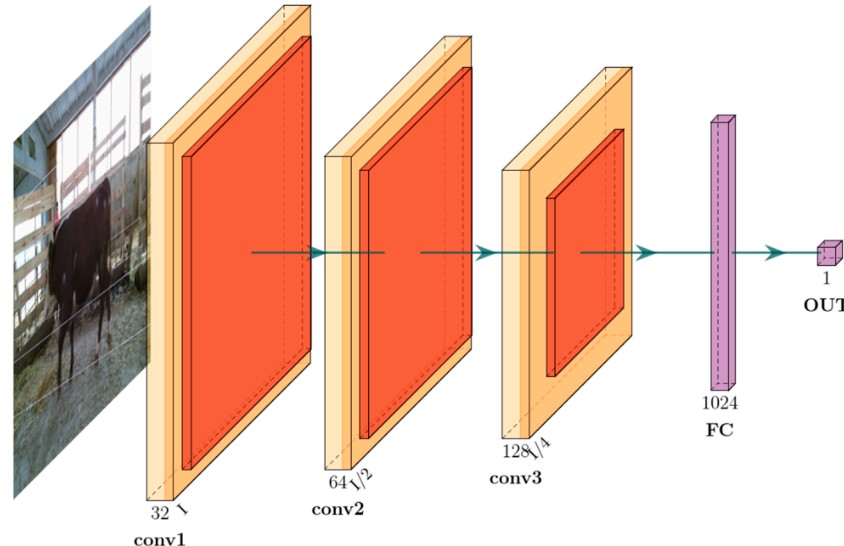

**Figure 11.** MRGB Convolutional Neural Network.

The model (MDM) for cow live weight prediction is shown in Figure 12, where only depth maps are input. The model contains 3 convolution blocks (con1, conv2, and conv3) and two fully connected layers (full and out). The conv1 block has 64, $3 \times 3$ filters, the conv2 block has 64, $3 \times 3$ filters, and the last convolution block, conv3, has 256, $3 \times 3$ filters. After the conv1, conv2, and conv3 blocks, maximum pooling layers with a kernel size of $2 \times 2$ are used.

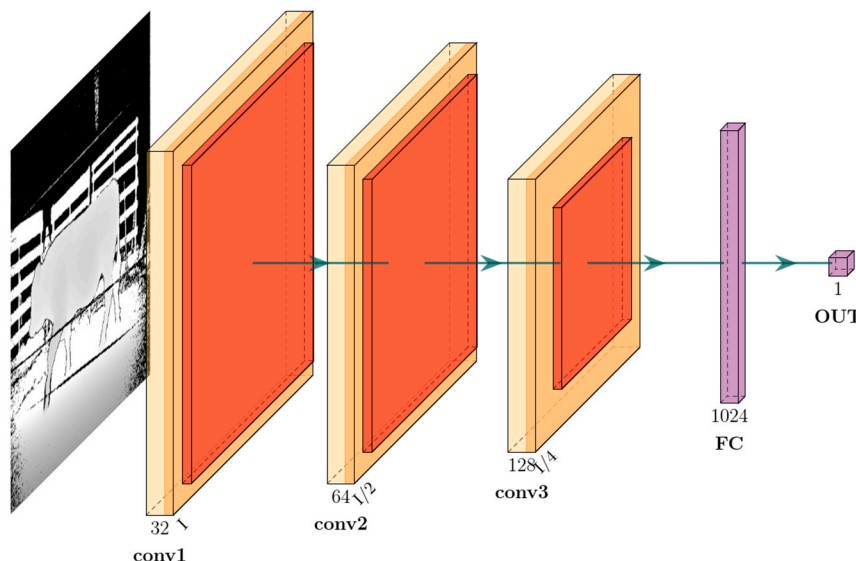

**Figure 12.** MDM Convolutional Neural Network.

The main goal of the proposed models is to estimate the live weight of a cow using an input RGB image or/and an input depth map. The live weight regression of the input image is calculated using loss functions as follows:

$$P_Y(\Theta) = \frac{1}{n} \sum_{i=1}^{n} \| F_y(X_i, \Theta) - Y_i \|^2, \tag{8}$$

where $\Theta$ is the set of parameters of the CNN model, $X_i$ is the input RGB image or/and the input depth map, and $n$ is the number of the training dataset. $P_Y$ is the loss between the estimated

weight $F_y(X_i; \Theta)$ (the output of the fully connected layer OUT) and the ground truth weight $Y_i$. The loss function is minimized using mini-batch gradient descent and backpropagation.

Initially, the dataset was randomly divided into training (70%) and test (30%) subsets. In addition, 20% of the training subset is used for validation of deep learning. Hyperparameter optimization is important for solving the problem of choosing the set of optimal hyperparameters. A traditional way to optimize hyperparameters using cross-validation search (GridSearchCV) is exploited, which is simply an exhaustive search over a manually specified subset of the deep network's hyperparameter space. With the help of GridSearchCV, optimal hyperparameters were found for all models used.

### 3.10. Data Augmentation

With a small number of images available, there is little variety in the data, which can lead to overfitting. To solve this problem, it is necessary to supplement the training data with synthesized and modified images. To supplement the data, a combination of rotation transformations along the three axes *X*, *Y*, and *Z* by $\pm 5$ degrees and a shift in height and width by $\pm 50$ cm was used, which corresponds to a shift along the *X* and *Y* axes, as well as random scaling in the range of 0.8m–1.2m, which is responsible for a shift along the *Z* axis. When using data argumentation, the total data size increased by 10 times. In total, the first complete dataset consists of 31,070 RGB images and 31,070 depth maps for 154 Hereford cattle. The second complete dataset consists of 28,630 RGB images and 28,630 depth maps for 121 Angus cattle.

### 3.11. Transfer Learning

Deep learning models require a lot of data to properly and accurately train. In the field of agriculture, it is difficult to obtain such large datasets due both to the limited number of studies carried out on a single farm and the amount of work required for manual measurements of animals. Therefore, the transfer learning approach is utilized. To do this, it is necessary to reuse pre-trained CNN models created for other tasks by retraining them for the problem. The pretrained EfficientNet model used in this paper has been previously used for image classification [39]. Transfer learning is used as a feature extractor, i.e., all layers are frozen, and only the top layer of the original classifier is retrained for new target classes. After that, it is necessary to set up a neural network to predict the live weight of the animal from RGB images and depth maps, or color projections and 2.5D projections. We carry over all the weights from EfficientNet but replace the last fully connected layer (FC8) with a new last fully connected layer and a softmax layer. The new last layer has a size of 1, and the weights are initialized randomly from a Gaussian distribution with a zero mean and a standard deviation of 0.01. A SGD with mini-batches of 32 samples is used, and a learning rate of 0.001 for the pretrained layers and a learning rate of 0.01 for the last output layer is set.

### 3.12. Performance Evaluation of Models

To evaluate the performance of the models, the mean absolute error (*MAE*) and mean absolute percentage error (*MAPE*) are used.

$$MAE = \frac{1}{n} \sum_{i=1}^{n} |y_i - f_i| \tag{9}$$

$$MAPE = \frac{1}{n} \sum_{i=1}^{n} \left| \frac{y_i - f_i}{y_i} \right| \tag{10}$$

where $n$ is a number of samples of the dataset, $\bar{y}$ is the mean of all known values of live weight, $y_i, i = 1, \ldots, n$ is a known value of live weight, and $f_i, i = 1, \ldots, n$ is a predicted value of live weight.

## 4. Results

We trained models on the Tesla V100 GPU for 27 epochs. The optimizer used was Adam. The learning, The batch size was set to the maximum possible within the allowable range of 16 GB of video memory. The model of each epoch was tested on the test dataset, and if the model had a lower *MAE* value (mean absolute error between the predicted weight and the true value) than on the training set, it was saved. There were 48,000 images in the training dataset and 11,700 images in the test dataset, with 123 Hereford and 96 Angus animals. The test dataset consisted of 31 Hereford and 25 Angus animals that were not associated with the training dataset.

The results showed that models get better when data augmentation and fine-tuning are used. In this work, in addition to RGB images and depth maps, color, and 2.5D projections are also used.

The experimental results are shown in Table 1. The original MRGBDM, MRGB, MDM networks, and EfficientNet (ENET) are trained, using different combinations of RGB images and depth maps as input, as well as RGB projections and depth map projections. Table 1 contains the final accuracy of cattle live weight prediction for each model tested. The best model is the proposed MRGBDM model, with a *MAPE* of 8.4 using RGB images and depth maps. It can be seen that the use of RGB projection and map projection can significantly reduce the *MAE* and *MAPE* errors. One can observe that the depth map contains many valuable features for image regression, in contrast to the RGB image. The performance of the pretrained ENET is worse compared with that of the MRGBDM and MRGB models.

**Table 1.** Live weight prediction results for cattle using the proposed MRGBDM, MRGB, and MDM models and the pre-trained EfficientNet (ENET) model on training and test datasets.

| Input to CNN | Model | Training Data | | | Test Data | | |
|---|---|---|---|---|---|---|---|
| | | *MAE* | *MAPE* | Accuracy | *MAE* | *MAPE* | Accuracy |
| Raw RGB images and depth maps | MRGBDM | 37.9 | 9.1 | 90.9 | 40.1 | 9.6 | 90.4 |
| | MRGB | 46.9 | 11.1 | 88.9 | 50.3 | 11.9 | 88.1 |
| | MDM | 40.5 | 9.5 | 90.5 | 43.5 | 10.2 | 89.8 |
| | ENET | 41.1 | 9.8 | 90.2 | 43.6 | 10.4 | 89.6 |
| Color and depth map projections | MRGBDM | 34.2 | 8.1 | 91.9 | 35.5 | 8.4 | 91.6 |
| | MRGB | 42.5 | 10.1 | 88.9 | 45.6 | 10.8 | 89.2 |
| | MDM | 37.6 | 8.9 | 91.1 | 39.7 | 9.4 | 90.6 |
| | ENET | 38.9 | 9.2 | 90.8 | 41.8 | 9.9 | 90.1 |

The results of this study show that the MRGBDM model can be used to predict the live weight of cattle. It is interesting to note that in a preliminary study [17], the conclusion was similar. In contrast, instead of using entire RGB images and depth maps, animal area selection from RGB-D images and 3D data augmentation are performed. This led to an increase in the accuracy of body weight prediction. The obtained results can help researchers and farmers implement deep learning algorithms for accurate body weight prediction using image regression. From our point of view, indirect automated live weight estimation should consist of non-invasive morphometric measurements based on computer vision, followed by body weight prediction using deep learning.

## 5. Conclusions

Non-contact weight measurement saves time and stress for cattle. The best option is to measure the weight from RGB images and depth maps. Two cattle datasets were used for experiments, that is, 154 Hereford and 121 Aberdeen Angus animals. To use deep neural networks, several problems need to be solved, such as environmental variability, noise, and missing parts of the data, the position in space of the animal, scaling, and a small sample of

training data. Therefore, algorithms for preprocessing RGB images and depth maps were designed, including noise suppression on RGB images and depth maps, background removal, and normalization of animal posture data. After processing, the data is fed into the weight prediction model. In this paper, it was proposed to use RGB projection and depth map projection as an input to a deep neural network instead of raw RGB images and depth maps, which made it possible to increase the reliability of cattle live weight prediction.

To predict the live weight of cattle, three models based on image regression were used. An evaluation of the performance of the proposed MRGBDM, MRGB, MDM, and pre-trained ENET models was obtained using neural network fine-tuning and data augmentation methods. The best model is the proposed MRGBDM model with the RGB and depth map projections. Experimental results on real data showed that the proposed MRGBDM model provides a high weight measurement accuracy of 91.6% (*MAPE* is 8.4).

Non-contact measurement of the live weight of cattle can be used in agriculture: for an objective assessment of breeding animals during grading; to assess the commercial value of livestock during the work of livestock auctions in different countries; to justify the further use of young animals, including for fattening, with the prospect of eliminating the need to perform a genetic examination of animals for the presence of genes; and to develop an analog technology for assessing the health and productivity of animals in industrial complexes.

In the future, to improve the accuracy of image regression, one can use a pre-processed point cloud as the input to a deep neural network. It is also desirable to expand the training database with new data, such as different breeds, age groups, and weight groups.

**Author Contributions:** Conceptualization, V.K. (Vitaly Kober) and A.R.; methodology, V.K. (Vladimir Kolpakov) and A.R.; software, A.R., A.G., and K.D.; validation, V.K. (Vladimir Kolpakov) and A.R.; formal analysis, A.R. and V.K. (Vitaly Kober); investigation, A.R., V.K. (Vladimir Kolpakov), and K.D.; resources, V.K. (Vladimir Kolpakov) and A.R.; data curation, A.R., A.G., and K.D.; writing—original draft preparation, V.K. (Vitaly Kober) and A.R.; writing—review and editing, H.G., V.K. (Vitaly Kober), and A.R.; visualization, A.R., A.G., and K.D.; supervision, A.R.; project administration, A.R. All authors have read and agreed to the published version of the manuscript.

**Funding:** The research was carried out with the financial support of the Russian Science Foundation within the framework of the scientific project No. 21-76-20014.

**Institutional Review Board Statement:** All animal protocols used in this study were approved by the institutional Animal Care and Use Committee of the Federal Research Centre of Biological Systems and Agro-technologies of the Russian Academy of Sciences (Russian).

**Informed Consent Statement:** Not applicable.

**Data Availability Statement:** The data presented in this study are available.

**Conflicts of Interest:** The authors declare no conflict of interest. The funders had no role in the design of the study; in the collection, analyses, or interpretation of data; in the writing of the manuscript, or in the decision to publish the results.

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
