# Peer review of "Live Weight Prediction of Cattle Based on Deep Regression of RGB-D Images"

_agriculture, doi:10.3390/agriculture12111794_

Round 1
Reviewer 1 Report
The manuscript is very well written and the study is sound. I have the following suggestions for improvement:
-The abstract should be supplemented with specific information of the study and should be standalone
-Line 24: Instead of 'peer reviewed' write 'reviewed'
-Line 51: Instead of [14] proposed write the name of the author. You may write 'author' and co-workers proposed...
-Line 150: Revise the statement with name of author instead of [25], similarly change Line 155, 157, 170, 186, 238
-Line 396 and 404: Make subheadings for image resize and signal range normalization
-MRGB, MRGBDM and MDM models have the same features hence, it is better to merge the discussion for these.
-Avoid the use of 'we', 'our' etc. in the paper
-The accuracy information of the models is missing in Table 1. Please supplement it.
-The discussion could be elaborated for better understanding of the results
-The challenges associated with this study may be added to the manuscript.
Author Response
Dear Reviewer 1,
Thank you for your valuable comments and suggestions.
The manuscript was revised according to your remarks. We did our best to improve the presentation of our results, but if you have further suggestions we will be pleased to implement them.
The details of changes and revisions are reported in the “Response to Reviewer 1 Comments” PDF file.
Sincerely,
Alexey Ruchay (On behalf of all co-authors)

Reviewer 2 Report
The paper proposes a new model for predicting live weight based on image regression with deep learning and efficient preprocessing of RGB images and depth maps, as well as the creation of color RGB projections and 2.5D depth maps for subsequent real-time weight prediction. It is an interesting topic to the researchers in the related areas. However, there still need some significant improvements. Here are some suggestions.
1. The format of the figure is not uniform in the overall structure, and the description of the picture is too simple. The description of the image should clearly express the content of the image, so as to better understand the paper. In addition, there is something wrong with the formatting of the table in the paper.
2. The authors give much description of Materials and Methods. However, the results part seems insufficient. The experiments were conducted on only one dataset. Moreover,
there is only one way to enhance the data. Are there more ways to enhance the data?
3. The number of times the model is trained is relatively small, which cannot make sure the reliability of the test result. In addition, there exist not experiments to prove the potential applicability of the proposed approach to animal husbandry.
4. Some related works should also cited ore discussed.
(1) Automatic Weight Prediction System for Korean Cattle Using Bayesian Ridge Algorithm on RGB-D Image
(2) Live weight prediction of cattle using deep image regression
(3) A Survey of Convolutional Neural Networks: Analysis, Applications, and Prospects
Author Response
Dear Reviewer 2,
Thank you for your valuable comments and suggestions.
The manuscript was revised according to your remarks. We did our best to improve the presentation of our results, but if you have further suggestions we will be pleased to implement them.
The details of changes and revisions are reported in the “Response to Reviewer 2 Comments” PDF file.
Sincerely,
Alexey Ruchay (On behalf of all co-authors)

Reviewer 3 Report
Comments:
1- Can you tell us how the objectives stated in the study were achieved? You can also add it to the introduction section as a short
2- State clearly what the main problem(s) are.
3- It is recommended to further enrich the related works. Definitely include the pros and cons of the methods discussed in the article.
4- Can you prove that your dataset is reliable? You can provide persuasive comments or references.
5- Equations written must have numbers.
6- It is recommended that you provide more detailed information on feature selection.
7- What are your hyperparameters and features? In addition, it is necessary to give information about tunning (ex:https://doi.org/10.1016/j.compbiolchem.2021.107619).
8- Figures 10, 11, and 12 are very similar to each other and the difference between them could not be highlighted in them. Revise.
9- After the results section or in the same section, add the discussion topic to the article.
10- The comparisons are quite inadequate. Compare your results with some methods in the literature.
11- Add your future works
12- Include the strengths and weaknesses of your work.
Author Response
Dear Reviewer 3,
Thank you for your valuable comments and suggestions.
The manuscript was revised according to your remarks. We did our best to improve the presentation of our results, but if you have further suggestions we will be pleased to implement them.
The details of changes and revisions are reported in the “Response to Reviewer 3 Comments” PDF file.
Sincerely,
Alexey Ruchay (On behalf of all co-authors)

Round 2
Reviewer 3 Report
Requested corrections and comments have been answered.